# The Assessment of Disability in Italy: The Laborious Procedure and Sharing of Objectives

**DOI:** 10.3390/ijerph192113777

**Published:** 2022-10-23

**Authors:** Giuseppe Consolazio

**Affiliations:** Medical Office, National Social Security Institute (INPS), 09016 Iglesias, Italy; giuseppe.consolazio@inps.it

**Keywords:** disability, civil invalidity, disability assessment

## Abstract

The assessment of disability in Italy requires the support of a system entirely dedicated to forensic evaluative medicine, which, for years, has been associated with the National Social Security Institute (INPS). Its medical offices are daily engaged in evaluating applications submitted by citizens. Their examination takes place in two different ways in the various Italian regions: assessments carried out by the Local Health Authority (ASL) and controlled by the INPS; evaluations carried out entirely by the INPS only. The main problem observed, and not yet resolved, is the excess time taken to respond to a citizen’s request, especially in areas where the procedure retains the biphasic ASL–INPS modality. This phenomenon is exemplified by the presentation of cases of the INPS medical office of Iglesias (South Sardinia, Italy), which include a series of disability applications examined in the year 2021 from January to September. The most favourable feedback is a tested and shared path in the determination of judgments.

## 1. Introduction

The assessment of disability in Italy is a complicated procedure, especially because, in most of the country, it consists of two phases based on the involvement and coordination of two institutions, the Local Health Authority (ASL) and National Social Security Institute (INPS). This article describes this procedure in Iglesias (South Sardinia), which reflects the modalities and times observable in most of the Italian regions. The assessment of disability in Italy is divided into five areas: civil invalidity (law no. 118/1971 and subsequent amendments), blindness (law no. 382/1970 and subsequent amendments), deafness (law no. 381/1970 and subsequent amendments), handicaps (law no. 104/1992), and the targeted job placement of disabled people (law no. 68/1999). Civil invalidity, which is determined by a commission of doctors only, provides for the granting of economic and non-economic benefits proportionate to the disability caused by all the health conditions (disorders and diseases), and it is measured based on three parameters: a permanent reduction in working capacity in subjects aged 18–65 years, derived from the percentage measures of health disorders and diseases indicated in the table provided by the Ministerial Decree of 5 February 1992; persistent difficulties in performing age-related tasks and functions in subjects under 18 years old and over 65 years old; dependence on third parties in walking and/or daily life in all age groups [1]. The civil invalidity of people of working age is expressed as percentage measures of the incapacity to work: 0–33%, not disabled; 34–73%, disabled with the right to non-economic benefits only (for example, targeted job placements for civil invalidity ≥46%); 74–100%, disabled with the right to economic benefits. The civil invalidity of minors/elderly is not linked to the percentage measurement of the diseases in the table and can only concern the two possibilities of granting or not granting an economic benefit. Independent financial compensation is guaranteed for disability caused by visual impairment [2] and prelingual deafness [3]. The normative discipline of the handicap, which is determined by a commission of doctors and social workers, identifies the nature of the handicap in a person who has a physical, psychological, or sensory impairment, stabilized or progressive, that is the cause of difficulties in learning, in relationships, or in work placements, to determine potential social disadvantages or marginalization. The nature of the disability for a handicap can be quantified in a binary fashion, as not severe or severe, according to whether it reduces personal autonomy and necessitates permanent and global assistance in the individual or relational sphere [4]. The recognition of a person with a disability qualified as a handicap entails the right to a wide range of mainly non-economic benefits (insertion and social integration, personal assistance services, school insertion and integration, various services, and paid absences from work for individuals with disabilities and their caregivers). The law on targeted job placements, the eligibility for which is determined by a commission of doctors and social workers, is aimed at promoting the job integration of disabled people including those in various categories, such as civil invalidity ≥46%, blindness, deafness, invalidity for work ≥33%, and receivers of social security for invalidity [5]. Every citizen can apply for the recognition of all five types of disabilities. There are often double applications for civil invalidity and handicaps, which have different purposes, the first being aimed at determining whether economic benefits should be granted and the second focused on the objective of inclusion and social integration. The rare occurrence of multiple and simultaneous economic requests for civil invalidity, blindness, and deafness can fall into the category of people with multiple disabilities, whose legal status, provided for by the sentence of the Constitutional Court no. 346/1989 and by art. 2 law no. 429/1991, admits the receipt of the various economic benefits [6]. In all areas of disability, each disease reported in the final diagnosis of each report is classified with the code of the International Classification of Diseases (ICD) [7]. The INPS plays a central role in the assessment of disability in Italy, in all the different ways of carrying out the procedure. The first-instance application for the required providence, accompanied by a preparatory medical certificate, is submitted online by the citizen to INPS. The operational flow takes place entirely online [8]. The evaluation of each instance, pursuant to article 20 law no. 102/2009, provides, in most of the national territory, a two-phase procedure: the first phase is an evaluation carried out by the ASL; the second phase is a final verification and evaluation carried out by INPS. In this manner, the visit is carried out by the ASL medical commission integrated by the INPS doctor (Integrated Medical Commission = IMC), and its opinion is subsequently verified by the competent INPS medical office (MO). The report drawn up by the IMC is validated by the MO or, in cases of disagreement, is suspended, and the evaluation of the application is repeated with a direct visit by the MO. The tacit consent rule, established by article 1 law no. 295/1990, dominates the procedure, so ASL opinions formulated by the IMC, that are not verified by the INPS MO within 60 days of their inclusion in the INPS database, are automatically confirmed. The INPS also carries out the assessment by itself from start to finish for first-instance visits in a limited part of the national territory, pursuant to article 18 law no. 111/2011, and for review visits throughout the national territory, pursuant to article 25 law no. 114/2014. Cases of disability applications are presented and examined in the year 2021 (January–September) at the INPS MO in Iglesias, in which the first-instance assessment procedure maintained the biphasic ASL–INPS organization.

## 2. Methods and Results

The operational flow of the disability assessment takes place completely online, collecting all the data coming from different and heterogeneous sources in a single dedicated online channel: the citizen’s application with the preparatory medical certificate attached, the report drawn up by the IMC, the final report of the MO, the archived medical records of each case. The data reported in this document were extracted from the INPS procedures for monitoring civil invalidity and by consulting the integrated database of civil invalidity (the term civil invalidity is used as part of the whole disability system). The chronology of the process is a sequence of moments in time (T) that identify the next steps (Figure 1).

The annual averages of the time intervals of the process (days) recorded in Iglesias in the years 2011–2021 are reported (Table 1 and Figure 2). The duration of the entire process (T0–T8) has always been very high; it showed a tendency to fall below the 200-day threshold only in the period 2016–2018, while it exceeded 300 days in 2011, 2014, and 2021.

The activity carried out by ASL–INPS coordination in the chosen period is summarized by the total number of applications, the ASL reports, the INPS reports, and the timing of the procedure (Table 2).

The ASL phase is represented by the distribution of the reports drawn up by the IMC and divided by the type of application (Table 3; Figure 3, Figure 4 and Figure 5).

Almost all the applications concerned civil invalidity and handicaps (Table 3; Figure 3).

The numbers of applications and ASL reports were almost the same (Table 3, Figure 4).

The average application processing time, equal to 360 days in the examined period of 2021, implied a range of values between 137 and 568 days for the different types of applications (Table 3; Figure 5).

The INPS phase is represented by the production of the final reports of the MO, resulting from the three main methods of processing the reports prepared by the IMC: validation, tacit consent, and direct visits (Table 4).

The results of the assessment in the civil invalidity area are presented, divided by age groups: subjects under 18 years old (Table 5), aged 18–65 years (Table 6), and over 65 years old (Table 7).

In subjects under 18 years old, the assessment of civil invalidity provides for three outcomes: not disabled, disabled minor with the right to the economic benefit of child disability allowance, or an accompanying allowance.

In subjects aged 18–65 years, the diversified percentage scale of civil invalidity has its turning point at 74%, which identifies the minimum level for granting economic benefits (74–99%—allowance; 100%—pension or pension plus accompanying allowance) and separates it from the lower levels, with a totally negative outcome (0–33%—not disabled) or eligible for only non-economic benefits (≥34%—prostheses; ≥46%—inclusion in a list of targeted work; ≥67%—exemption from medical expenses).

In subjects over 65 years old, the outcome of the assessment for civil invalidity is summarized with the two possibilities of granting/not granting the accompanying allowance.

Finally, a breakdown of the results for civil invalidity regarding economic and non-economic benefits is presented (Table 8).

## 3. Discussion

The participation of the INPS doctor in the visit carried out by the IMC responds to a twofold need: the sharing of the evaluation process between INPS and ASL doctors; control of the entire procedure. This element characterizes and qualifies the disability assessment procedure outlined by art. 20 law n. 102/2009 [9]. The sharing of the evaluation process relates to the perspective of collegiality that already distinguishes clinical medicine [10] and extends, by analogy, to disability assessment. The control action is carried out during the visit with the proposal by the INPS doctor of an opinion on the judgment, which can be shared or not. It continues with the subsequent definitive evaluation, which involves the examination of the entire medical record accompanying the ASL report and ends with its definitive validation or its suspension and the carrying out of a direct visit to the MO. The ASL–INPS procedure is an obstacle course and means that responding to citizens’ demands takes a very long time. The average time across all applications, which has remained high since 2011 in the history of the disability assessment procedure pursuant to art. 20 l. n. 102/2009 of Iglesias, peaked at 360 days in 2021; the times were different for different types of disabilities, reaching the extreme number of 568 days for targeted job placement applications. The long wait for the outcome of an application is caused by the sum of the multiple phases and stops within the ASL–INPS procedure. In this context, the main reason for the delay is the excessive waiting times for ASL visits (T1–T2). The reduction in health services caused by the COVID-19 pandemic [11] contributed to this further delay. The legal provisions stating that the visits must take place within 30 days of the date of the request and, for cancer patients, within 15 days, have never been met in Iglesias. These data are very disheartening, especially if compared with other regional realities such as the one reported in Modena, where, in the presence of a biphasic ASL–INPS procedure, a new organized protocol allowed visits within 15 days of the date of the submission of the application in 100% of cases [12]. Even the desire to guarantee the granting of a recognized economic benefit within 120 days of the date of submission of the application has never been significantly realized, with this occurring in only up to 30% of applications. In Iglesias, the INPS doctor was unable to attend all the IMC visits, and the control function was mainly carried out by the definitive assessment phase at the MO. The number of ASL reports suspended due to disagreement followed by a direct visit was irrelevant in 2021. Excluding the minority fraction of the reports removed from control by tacit consent, almost all the ASL reports were confirmed and validated by INPS. An appreciable and generalized agreement emerges between the decisions expressed by the IMC and the final judgment decreed by the MO, within the framework of a shared medico-legal perspective that unites all the operators involved. This statement is also confirmed by the 2021 data of the specific area of civil invalidity. The granting of economic benefits was clearly greater (70.9%) than their denial (29.1%). The highest grant rate was recorded in subjects under 18 years old (81%), followed by those over 65 years old (72.4%) and aged 18–65 years (67.1%). The data for subjects under 18 years old confirm the global finding that mental disorders are the leading cause of disability in young people [13]. Data from other age groups showed that the primary causes of disability were neoplasms [14] in adults and multimorbidity [15] in the elderly.

## 4. Conclusions

The assessment of disability is a medico-legal process whose correctness and fairness draw strength from the objective evidence of the ascertained and documented clinical–functional picture. Evaluators (medical and non-medical) must judge the pathologies and conditions of applicants to determine their eligibility or non-eligibility for benefits. This determination arises from the qualitative and quantitative identification of impairments in the areas of civil invalidity, blindness, and deafness according to the indications of the medical model [16]. The medico-legal evaluation of the disabling state in the context of civil invalidity must include the specific entity of the health disorder, the incidence of symptoms, and all effects on activities of personal and relational life, as well as quality of life, the response to therapies and their side effects, and comorbidities [17]. The decision-making mechanism extends to the consideration of socio-environmental factors in the areas of handicaps and targeted job placement. The problem not yet resolved in Iglesias, even in 2021, is the excessive delays in answering citizens’ questions. In some Italian regions (which still represent a minority of national territory), the disability assessment carried out only by INPS has normalized waiting times. Likewise, in the autonomous province of Trento, pursuant to provincial law no. 7/1998, the assessment of disability is carried out only by the ASL, without subsequent INPS verification, with standard waiting times [18]. Following the acknowledgement of the limitations and drawbacks of its slowness, the evaluation process adopted by the ASL and the INPS of Iglesias, in the described framework of 2021, appeared linear, uniform, and facilitated by a substantial agreement of opinions guaranteeing the rights of applicant citizens. Compared to the multifaceted biopsychosocial paradigm adopted by the International Classification of Functioning, Disability and Health (ICF) [19], the narrower medical model, focused on disease-related impairment, continues to provide, in the procedure described above, the minimum tool necessary to sufficiently verify the requirements of law for the type of disability under consideration. The disability assessment system in Italy appears destined to be radically reformed by law no. 227/2021 with the adoption of ICF.

## Figures and Tables

**Figure 1 ijerph-19-13777-f001:**
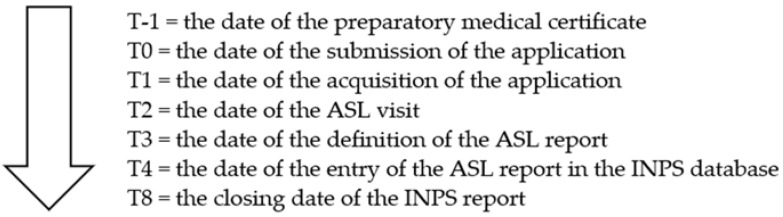
Dates of the disability assessment process.

**Figure 2 ijerph-19-13777-f002:**
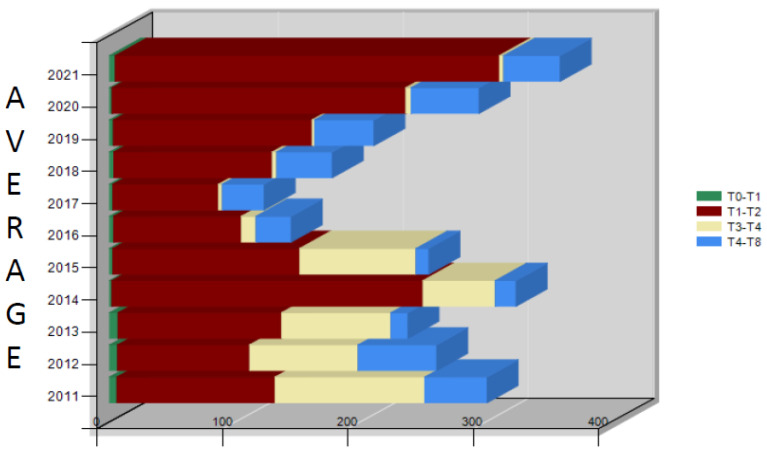
Iglesias: annual averages of the time intervals of the process (days) for 2011–2021.

**Figure 3 ijerph-19-13777-f003:**
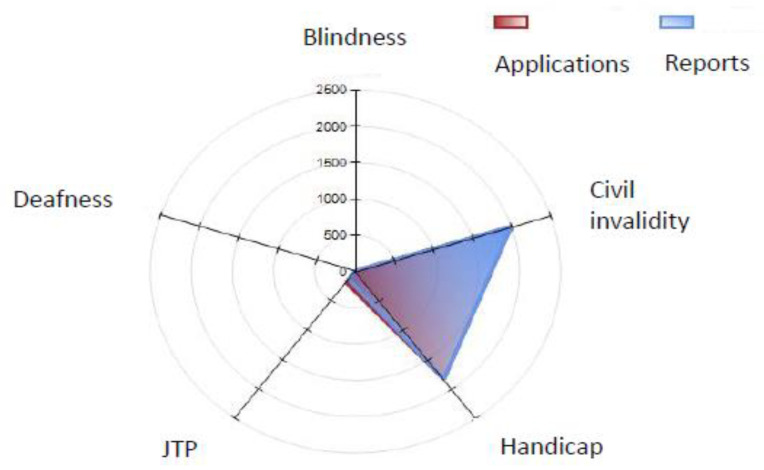
Iglesias: distribution of total applications and ASL reports for 2021 (January–September).

**Figure 4 ijerph-19-13777-f004:**
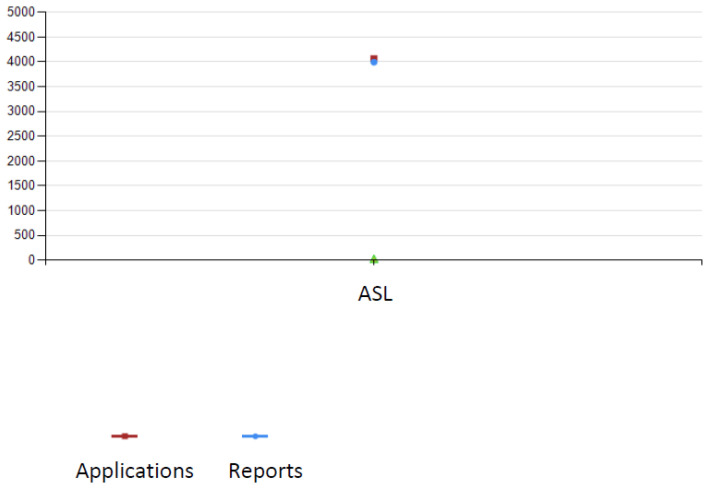
Iglesias: trend of total applications and ASL reports for 2021 (January–September).

**Figure 5 ijerph-19-13777-f005:**
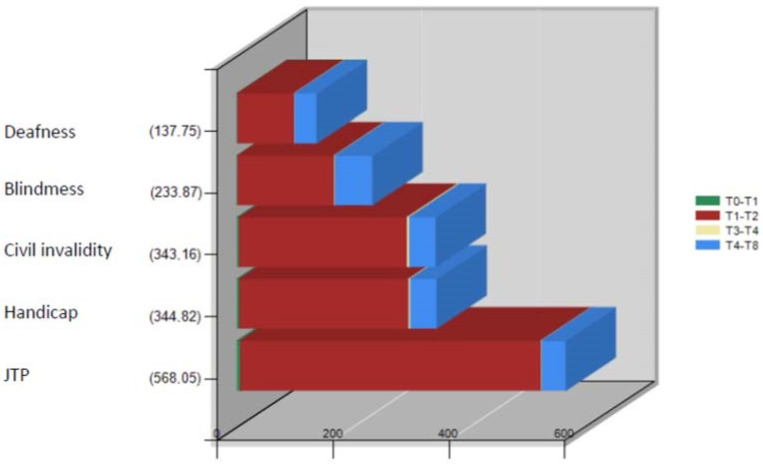
Trend in application processing time (days) for 2021 (January–September).

**Table 1 ijerph-19-13777-t001:** Iglesias: annual averages of the time intervals of the process (days) 2011–2021.

Average	T0–T1	T1–T2	T3–T4	T4–T8	T0–T8
2021	5	307	3	45	360
2020	2	235	4	54	295
2019	3	159	2	47	211
2018	3	127	4	44	178
2017	3	84	3	33	123
2016	3	102	12	28	145
2015	3	149	93	10	255
2014	2	248	58	16	325
2013	7	131	87	14	238
2012	6	105	87	63	261
2011	6	126	119	50	301

**Table 2 ijerph-19-13777-t002:** Iglesias summary data: applications, ASL and INPS reports, and average times of procedure.

ASL-INPS	2021 (January–September)							
				Average Times (Days)				
	Applications	ASL Reports	INPS Reports	T0–T1	T1–T2	T3–T4	T4–T8	T0–T8
	4063	3994	4373	5	307	3	45	360

**Table 3 ijerph-19-13777-t003:** Iglesias: distribution of ASL reports by type of application.

ASL	2021 (January–September)					
	Blindness	Civil Invalidity	Handicap	JTP	Deafness	Total
Applications	22	1990	1846	197	8	4063
Reports	23	2001	1830	136	4	3994
T1–T2 (days)	167	302	297	535	98	307

JTP = Job Targeted Placement.

**Table 4 ijerph-19-13777-t004:** Iglesias: distribution of INPS reports by the type of application and ASL report processing in 2021 (January–September).

INPS	Blindness			Civil Invalidity			Handicap	JTP	Deafness			Total		
ASL R	NEB	EB	T	NEB	EB	T			NEB	EB	T	NEB	EB	T
VAL	6	14	20	573	1392	1965	1715	120	5	0	5	2419	1406	3825
TC	0	6	6	94	226	320	192	21	1	0	1	308	232	540
DV	2	2	4	0	4	4	0	0	0	0	0	2	6	8
T	8	22	30	667	1622	2289	1907	141	6	0	6	2729	1644	4373

ASL R = ASL report. NEB = non-economic benefit. EB = economic benefit. VAL = validated. TC = tacit consent. DV = direct visit. T = total. JTP = job targeted placement.

**Table 5 ijerph-19-13777-t005:** Iglesias: civil invalidity in subjects under 18 years old in 2021 (January–September).

	Males: Mean Age, 9 ± 3.83				Females: Mean Age, 9.65 ± 4.21				All: Mean Age, 9.28 ± 4			
DISEASES	ND	CDA	AA	T	ND	CDA	AA	T	ND	CDA	AA	T
Total	18	68	10	96	14	52	6	72	32	120	16	168
Mental	15	54	8	77	13	40	2	55	28	94	10	132
Inborn	2	2	2	6	1	3	4	8	3	5	6	14
Diabetes	0	7	0	7	0	6	0	6	0	13	0	13
Other	1	5	0	6	0	3	0	3	1	8	0	9

ND = not disabled. CDA = child disability allowance. AA = accompanying allowance. T = total.

**Table 6 ijerph-19-13777-t006:** Iglesias: civil invalidity in subjects aged 18–65 years in 2021 (January–September).

	Males: Mean Age, 53.33 ± 11.6				Females: Mean Age, 51.96 ± 11.22				All: Mean Age, 52.63 ± 11.43			
DISEASES	NEB	A/P	PAA	T	NEB	A/P	PAA	T	NEB	A/P	PAA	T
Total	169	148	143	460	140	180	160	480	309	328	303	940
Neoplasms	5	27	78	110	15	48	110	173	20	75	188	283
Articular	58	9	0	67	39	19	2	60	97	28	2	127
Mental	9	24	25	58	17	32	18	67	26	56	43	125
Multiple	11	25	10	46	16	33	6	55	27	58	16	101
Nervous	7	7	15	29	8	16	16	40	15	23	31	69
Diabetes	27	13	3	43	11	12	2	25	38	25	5	68
Cardiac	26	22	2	50	6	4	1	11	32	26	3	61
Respiratory	14	8	2	24	7	0	0	7	21	8	2	31
Immune	3	1	1	5	11	8	3	22	14	9	4	27
Digestive	3	8	3	14	1	5	0	6	4	13	3	20
Genitourinary	0	2	2	4	6	1	2	9	6	3	4	13
Inborn	1	1	2	4	2	2	0	4	3	3	2	8
Ocular	3	1	0	4	0	0	0	0	3	1	0	4
Hearing	2	0	0	2	1	0	0	1	3	0	0	3

NEB = non-economic benefit (not disabled; partially disabled, 34–73%). A/P = allowance/pension (partially disabled, 74–99% — allowance; totally disabled, 100% — pension). PAA = pension (totally disabled, 100%) plus accompanying allowance. T = total.

**Table 7 ijerph-19-13777-t007:** Iglesias: civil invalidity in subjects over 65 years old in 2021 (January–September).

	Males: Mean Age, 79.69 ± 8.11			Females: Mean Age, 81.28 ± 7.88			All: Mean Age, 80.65 ± 8.01		
DISEASES	NAA	AA	T	NAA	AA	T	NAA	AA	T
Total	103	362	465	223	493	716	326	855	1181
Multiple	22	68	90	49	130	179	71	198	269
Neoplasms	19	126	145	30	71	101	49	197	246
Dementia	3	66	69	25	136	161	28	202	230
Locomotive	16	19	35	65	85	150	81	104	185
Cardiac	22	20	42	23	23	46	45	43	88
Nervous	1	36	37	6	38	44	7	74	81
Diabetes	8	8	16	14	3	17	22	11	33
Respiratory	9	5	14	6	3	9	15	8	23
Dialysis	0	8	8	1	3	4	1	11	12
Cirrhosis	2	4	6	1	1	2	3	5	8
Ocular	1	2	3	2	0	2	3	2	5
Hearing	0	0	0	1	0	1	1	0	1

NAA = no accompanying allowance. AA = accompanying allowance. T = total.

**Table 8 ijerph-19-13777-t008:** Iglesias: economic benefits for civil invalidity or not in 2021 (January–September).

Age	Diseases	NEB No (%)	EB No (%)	Total No (%)
ALL	Total	667 (29.1)	1622 (70.9)	2289 (100)
<18	Total	32 (19)	136 (81)	168 (100)
	Mental	28 (21.2)	104 (78.8)	132 (78.6)
	Inborn	3 (21.4)	11 (78.6)	14 (8.3)
	Diabetes	0 (0)	13 (100)	13 (7.7)
	Other	1 (11.1)	8 (88.9)	9 (5.4)
18–65	Total	309 (32.9)	631 (67.1)	940 (100)
	Neoplasms	20 (7.1)	263 (92.9)	283 (30.1)
	Articular	97 (76.4)	30 (23.6)	127 (13.5)
	Mental	26 (20.8)	99 (79.2)	125 (13.3)
	Multiple	27 (26.7)	74 (73.3)	101 (10.7)
	Nervous	15 (21.7)	54 (78.3)	69 (7.3)
	Diabetes	38 (55.9)	30 (44.1)	68 (7.2)
	Cardiac	32 (52.5)	29 (47.5)	61 (6.5)
	Respiratory	21 (67.7)	10 (32.3)	31 (3.3)
	Immune	14 (51.9)	13 (48.1)	27 (2.9)
	Digestive	4 (20)	16 (80)	20 (2.2)
	Genitourinary	6 (46.2)	7 (53.8)	13 (1.4)
	Inborn	3 (37.5)	5 (62.5)	8 (0.9)
	Ocular	3 (75)	1 (25)	4 (0.4)
	Hearing	3 (100)	0 (0)	3 (0.3)
>65	Total	326 (27.6)	855 (72.4)	1181 (100)
	Multiple	71 (26.4)	198 (73.6)	269 (22.8)
	Neoplasms	49 (19.9)	197 (80.1)	246 (20.8)
	Dementia	28 (12.2)	202 (87.8)	230 (19.5)
	Locomotive	81 (43.8)	104 (56.2)	185 (15.7)
	Cardiac	45 (51.1)	43 (48.9)	88 (7.5)
	Nervous	7 (8.6)	74 (91.4)	81 (6.9)
	Diabetes	22 (66.7)	11 (33.3)	33 (2.8)
	Respiratory	15 (65.2)	8 (34.8)	23 (1.9)
	Dialysis	1 (8.3)	11 (91.7)	12 (1)
	Cirrhosis	3 (37.5)	5 (62.5)	8 (0.6)
	Ocular	3 (60)	2 (40)	5 (0.4)
	Hearing	1 (100)	0 (0)	1 (0.1)

NEB = non-economic benefit. EB = economic benefit.

## Data Availability

The data reported in this document were extracted from the INPS procedures for monitoring civil invalidity and by consulting the integrated database of civil invalidity accessible from http://intranet.inps.it/PORT01/Intranet/Portale/FrmTemplateFrame.aspx.

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
