# Peer review of "The Assessment of Disability in Italy: The Laborious Procedure and Sharing of Objectives"

_ijerph, 2022, doi:10.3390/ijerph192113777_

Round 1

Reviewer 1 Report

Thank you for this interesting paper.  People do not understand how governments process disability claims. It would be useful to have a paper on this process in an open source paper.   

Now that being said, this paper would benefit from English language editing. Grammatically it is mainly fine, however, the readability can be improved. For example where explanations are placed in sentences and the word choice makes this paper much  more difficult for readers to understand.

As well please check Purdue Owl for paragraphs in academic writing https://owl.purdue.edu/owl/general_writing/academic_writing/paragraphs_and_paragraphing/index.html.  Paragraphs improve the readability of a paper.  Your discussion is one paragraph. It needs to be separated into paragraphs.

Introduction

What terribly stigmatizing names for disability.  I realize that is not the author’s responsibility.

“ The assessment of disability in Italy is divided into five sectors: civil invalidity (law 20 no. 118/1971 and subsequent amendments), blindness (law no. 382/1970 and subsequent 21 amendments), deafness (law no. 381/1970 and subsequent amendments), handicap (law 22 no. 104/1992), job targeted placement of disabled people (law no. 68/1999).”

I realize these are government documents that are 23 to 53 years old, but worldwide we did have a very active disability movement in the 1960’s before these terms were coined. The World Health Organization certainly has better terms!

The author’s use of the words “invalidity” and “infirmities”  and “infirmity”:   Just because the government uses these derogatory terms, does not mean that you as the author need to use these terms.  Please use the WHO International Classification of Functioning, Disability and Health (ICF) terms https://www.who.int/standards/classifications/international-classification-of-functioning-disability-and-health

Nice definition of each category of disability and well explained. In this sentence, “The tacit consent rule, established 51 by article 1 law no. 295/1990, dominates the procedure so that the ASL reports not verified 52 within 60 days of their inclusion in the INPS database automatically become definitive.” “Definitive” is an unusual word to use in English. Does it mean that the application is accepted or rejected?  The way you use it does not inform the reader of any disposition.

MDPI references are usually at the end of the paper rather than in footnotes.

Results

You have provided data in tables and figures without any explanation. This is unusual. Typically, the author does not expect the reader to interpret the data, rather they explain what they want the reader to notice in the text. The tables and figures complement the text they are not the way findings are reported. Your discussion, more if not all, should be moved to results.

Discussion

The discussion should be a series of paragraphs in which you discuss your results with other literature. Typically the author summarizes their findings in a couple of sentences in the first paragraph and then discusses them in relation to OTHER published academic literature or reports.  Is there any Italian literature about the disability assessment process? What are the strengths and weaknesses of the process? What do people with disabilities say?

Take this paper back to the drawing board, and rewrite it in the academic paper style. See Purdue writing center for resources.  If you are looking for a good book on disability studies, See some of Albrecht’s work[1,2], or Albrecht,  Seelman, Bury, Handbook of Disability Studies. It is an old book (2001) but younger than the disability legislation.

References

1.         Albrecht, G.L.; Devlieger, P.J. The disability paradox: High quality of life against all odds. Social Science and Medicine 1999, 48, 977-988, doi:10.1016/S0277-9536(98)00411-0.

2.         Albrecht , H.; Comartin, J.; Valeriote, F. Not to Be Forgotten: Care of Vulnerable Canadians Final report of the ad hoc Parliamentary Committee on Palliative and Compassionate Care. 2010 November

Author Response

As suggested, the article benefited from English language editing.

The term civil invalidity identifies a category of disability provided for by law and cannot be replaced. The term infirmity has been replaced by health conditions (disorders and diseases).

Thanks for the comment "Nice definition of each category of disability and well explained". - The sentence on tacit consent has been corrected in the revised file => “The tacit consent rule, established by article 1 law no. 295/1990, dominates the procedure, so ASL opinions formulated by the IMC, that are not verified by the INPS MO within 60 days of their inclusion in the INPS database, are automatically confirmed”.

The explanation of the figures and tables has also been included in cases where it was lacking or absent. - There is little Italian literature on the subject in question (disability assessment process). In the revised file I have added 5 articles to the bibliography from the Italian Review of Legal Medicine. These articles, as well as others not cited, mainly address issues related to the medico-legal assessment methodology.

One, published in 2011, at the beginning of the civil invalidity procedure governed by art. 20 law 102/2009, was specifically dedicated to the issue of the disability assessment process.

One, published in 2018, should be underlined as it documented a reality opposite to that described for the demonstrated timeliness of the answers to the questions of cancer patients.

The request for the opinion of the disabled is a very interesting indication for a new study, but it goes beyond the scope of the study in question, which, about the strengths and weaknesses of the process, according to what was announced in the title of the revised file (The assessment of disability in Italy: the laborious procedure and sharing of objectives), has documented how the obstacles and difficulties inherent in the described, laborious procedure have not prevented the two institutions involved, ASL and INPS, from building a shared medico-legal path, aimed at protecting the rights of citizens.

Reviewer 2 Report

While the overarching topic of disability assessment is of great significance, this paper requires extensive improvements. 

First, the introduction provides an overview of the Italian assessment system but does not engage with the academic literature on the topic more broadly. Furthermore, the author included no research aim or question, so it is unclear to the reader what the paper's purpose or contribution is.

Second, the methods and results are combined. There is no description of how the data was acquired or analysed. The results are a series of 8 tables and five figures with very little accompanying text. 

Third, the discussion is primarily descriptive and does not engage with the disability assessment literature. 

I think there is potential for this type of data to contribute to the disability assessment literature, but the paper in its current form seems incomplete.

Author Response

The overview of the Italian disability assessment system is the main topic of the introduction, essential to understand the mechanisms and reasons for the problems addressed in the discussion. There is little Italian literature on the subject in question (disability assessment process). In the revised file I have added 5 articles to the bibliography from the Italian Review of Legal Medicine.

These articles, as well as others not cited, mainly address issues related to the medico-legal assessment methodology.

One, published in 2011, at the beginning of the civil invalidity procedure governed by art. 20 law 102/2009, was specifically dedicated to the issue of the disability assessment process.

One, published in 2018, should be underlined as it documented a reality opposite to that described for the demonstrated timeliness of the answers to the questions of cancer patients. 

The paper's purpose or contribution is now indicated in the first sentences of the introduction of revised file. “The assessment of disability in Italy is a complicated procedure, especially because, in most of the country, it consists of two phases, based on the involvement and coordination of two institutions, the Local Health Authority (ASL) and National Social Security Institute (INPS). This article describes this procedure in Iglesias (South Sardinia), which reflects the modalities and times observable in most of the Italian regions”. 

About how the data were acquired, the revised file adds that “The data reported in this document were extracted from the INPS procedures for monitoring civil invalidity and by consulting the integrated database of civil invalidity (the term civil invalidity is used as part of the whole disability system)”.

It is necessary to specify the work done of an observational study. The sum of the data provided by the INPS statistical procedures obviously required a subsequent processing and classification that emerges from the type of tables. The raw data summarized were the number of questions submitted by citizens, the number of responses provided by the responsible institutions, the times (tables 1-4 and figures 1-5). The subsequent series of tables 5-8, with additional accompanying text in the revised file, required a medico-legal classification and involved an epidemiological examination of the evaluated population.

The revised file increased the accompanying text of tables and figures.

The study, according to what was announced in the title of the revised file (The assessment of disability in Italy: the laborious procedure and sharing of objectives), has documented how the obstacles and difficulties inherent in the described, laborious procedure have not prevented the two institutions involved, ASL and INPS, from building a shared medico-legal path, aimed at protecting the rights of citizens.

Reviewer 3 Report

1) The introduction is very confusing.  I also think there is a language issue, as the word "sector" does not seem appropriate, unless I am missing something. Also it isn't clear if these five ways of getting disability status are mutually exclusive -- and if someone is both blind and has severe functional difficulties do they get to choose which way they get disability status? Can they qualify in multiple ways and get different types of benefits? It is all very unclear until you start reading the next section. I think it would be better to start with a brief paragraph that gives a general overview of the system. And also we go to "methods and results" before I have any idea what issue is being addressed. What is the research question? I have to infer it from the next section. Why not start off, for example: Italy has a single application process for disability benefits, that has five different application types leading to different statuses associated with different types of benefits. This paper will....

2) Also what is the discretion that people have in how they apply? If I am deaf can I apply for civil invalidity? If so, why would I choose one way over another? The system needs to be explained better. More importantly, who decides if I apply for "handicap" or "civil invalidity" and why? What are the tradeoffs in benefits? Is there a difference in the likelihood of being granted status?  All this should be explained before you start looking at the data.

3) I would not just look at the mean waiting times but also the medians. A large mean could be driven by a few big outliers. Besides, it would be interesting to know not just the mean wait time but the variance in wait times

4) Why are there such large delays? The paper would benefit from some in-depth interviews with stakeholders -- people in the system and people applying for benefits -- to learn more about their experiences to uncover the bottlenecks in the system.

This is a useful topic, but the paper is not adequately developed. It requires significant revision -- Just reporting on average waiting times is not a significant contribution. They need to explain the system, then document the nature of those waiting times -- do they differ by type of disability, for example -- and then gather some evidence on the factors creating those barriers.

Author Response

1) The revised file hopes to answer the reported problem of language and very confusing introduction.

The word “sector” has been replaced with “area”: “The assessment of disability in Italy is divided into five areas”.

The introduction of the revised file better specifies the various categories of disabilities within the space limits of an article. The question about multiple requests was addressed as follows: “Every citizen can apply for the recognition of all five types of disabilities. There are often double applications for civil invalidity and handicaps, which have different purposes, the first being aimed at determining whether economic benefits should be granted, and the second focused on the objective of inclusion and social integration. The rare occurrence of multiple and simultaneous economic requests for civil invalidity, blindness, and deafness can fall into the category of people with multiple disabilities, whose legal status, provided for by the sentence of the Constitutional Court no. 346/1989 and by art. 2 law no. 429/1991, admits the receipt of the various economic benefits”.

The research question is in the first sentences of the introduction: “The assessment of disability in Italy is a complicated procedure, especially because, in most of the country, it consists of two phases, based on the involvement and coordination of two institutions, the Local Health Authority (ASL) and National Social Security Institute (INPS). This article describes this procedure in Iglesias (South Sardinia), which reflects the modalities and times observable in most of the Italian regions”. 

2). As specified in the revised file, the data were extracted from the INPS procedure for monitoring civil invalidity which provided from thousands of data only the averages of waiting times, not the median and variance in wait times. 

3) Carrying out in-depth interviews with stakeholders - people of the system and people requesting benefits - is a very interesting indication for a new study, but it goes beyond the scope of the study in question which, according to what was announced in the title of the revised file (The assessment of disability in Italy: the laborious procedure and sharing of objectives), has documented how the obstacles and difficulties inherent in the described, laborious procedure have not prevented the two institutions involved, ASL and INPS, from building a shared medico-legal path, aimed at protecting the rights of citizens. The difference in waiting times for the different types of disabilities is indicated in the article (table 3 and figure

4). It’s clearly stated that the cause of the long waiting times lies in the biphasic ASL-INPS procedure. I propose again some phrases from the discussion: “The ASL–INPS procedure is an obstacle course and means that responding to citizens' demands takes a very long time. The average time across all applications, which has remained high since 2011 in the history of the disability assessment procedure pursuant to art. 20 l. n. 102/2009 of Iglesias, peaked at 360 days in 2021; the times were different for different types of disabilities, reaching the extreme number of 568 days for targeted job placement applications. The long wait for the outcome of an application is caused by the sum of the multiple phases and stops within the ASL–INPS procedure. In this context, the main reason for the delay is the excessive waiting times for ASL visits (T1–T2)”.

The counter-proof is found in a passage of the conclusions: “In some Italian regions (which still represent a minority of national territory), the disability assessment carried out only by INPS has normalized waiting times. Likewise, in the autonomous province of Trento, pursuant to provincial law no. 7/1998, the assessment of disability is carried out only by the ASL, without subsequent INPS verification, with standard waiting times”.

Author Response

New title of the revised file =>The assessment of disability in Italy: the laborious procedure and sharing of objectives

In my opinion, the abstract exposes the main contents of the article in an exhaustive way with a few sentences:

  • the articulated and diversified overview of the Italian disability assessment system; the most significant problem (long waiting times) of the system;
  • the appreciable evidence of a consolidated and shared medico-legal evaluation approach.

The study, according to what was announced in the abstract and in the title of the revised file (The assessment of disability in Italy: the laborious procedure and sharing of objectives), has documented how the obstacles and difficulties inherent in the described, laborious procedure have not prevented the two institutions involved, ASL and INPS, from building a shared medico-legal path, aimed at protecting the rights of citizens.

The paper's purpose or contribution is now indicated in the first sentences of the introduction of revised file. “The assessment of disability in Italy is a complicated procedure, especially because, in most of the country, it consists of two phases, based on the involvement and coordination of two institutions, the Local Health Authority (ASL) and National Social Security Institute (INPS).

This article describes this procedure in Iglesias (South Sardinia), which reflects the modalities and times observable in most of the Italian regions”. - The revised file adds that “The data reported in this document were extracted from the INPS procedures for monitoring civil invalidity and by consulting the integrated database of civil invalidity (the term civil invalidity is used as part of the whole disability system)”. It is necessary to specify the work done of an observational study.

The sum of the data provided by the INPS statistical procedures obviously required a subsequent processing and classification that emerges from the type of tables. The raw data summarized were the number of questions submitted by citizens, the number of responses provided by the responsible institutions, the times (tables 1-4 and figures 1-5).

The subsequent series of tables 5-8, with additional accompanying text in the revised file, required a medico-legal classification and involved an epidemiological examination of the evaluated population. 

The situation described in Iglesias is not a local specificity, it applies to most of the country (see the first sentences of the introduction). Problem about the word “The unified assessment of disability” ==correction==> “In some Italian regions (which still represent a minority of national territory), the disability assessment carried out only by INPS has normalized waiting times”.

Round 2

Reviewer 1 Report

Thank you. You have addressed my concerns. 

Reviewer 3 Report

Thanks for the edits. I believe they have made the paper much clearer. I recommended to accept.

Reviewer 4 Report

The author corrected the text as suggested. I have no comments.